# HubNet: An E2E Model for Wheel Hub Text Detection and Recognition Using Global and Local Features

**DOI:** 10.3390/s24196183

**Published:** 2024-09-24

**Authors:** Yue Zeng, Cai Meng

**Affiliations:** Image Processing Center, School of Astronautics, Beihang University, Beijing 100191, China; 20375077@buaa.edu.cn

**Keywords:** deep learning, wheel hub text, text detection, text recognition

## Abstract

Automatic detection and recognition of wheel hub text, which can boost the efficiency and accuracy of product information recording, are undermined by the obscurity and orientation variability of text on wheel hubs. To address these issues, this paper constructs a wheel hub text dataset and proposes a wheel hub text detection and recognition model called HubNet. The dataset captured images on real industrial production line scenes, including 446 images, 934 word instances, and 2947 character instances. HubNet is an end-to-end text detection and recognition model, not only comprising conventional detection and recognition heads but also incorporating a feature cross-fusion module, which improves the accuracy of recognizing wheel hub texts by utilizing both global and local features. Experimental results show that on the wheel hub text dataset, the HubNet achieves an accuracy of 86.5%, a recall of 79.4%, and an F1-score of 0.828, and the feature cross-fusion module increases the accuracy by 2% to 4%. The wheel hub dataset and the HubNet offer a significant reference for automatic detection and recognition of wheel hub text.

## 1. Introduction

Wheel hub, as a typical industrial die-casting component, is often marked with Latin characters and Arabic numerals using character embossing and laser marking techniques during the production process. These characters are arranged either transversely or in a circular pattern on the metal surface of the hub, recording critical information such as production serial number, date, part category, material, and process dimensions [1]. In actual production, due to the vast quantity of products, manually recording casting codes is inefficient, and errors are prone to occur over long periods of work.

The research on scene text detection and recognition tasks has a long history [2]. However, compared with normal scene texts, hub texts have low visual contrast, susceptibility to surface corrosion, and complex backgrounds with diverse orientations of the text. These characteristics significantly increase the difficulty of hub text detection and recognition. Therefore, existing text detection and recognition methods [3,4,5,6], which are designed to deal with natural scene text, have poor performances when directly applied to wheel hub text.

Current methods applied to text detection and recognition can primarily be categorized into traditional methods and deep learning methods. Traditional methods [7,8] often employ template matching algorithms for the identification and detection of cast characters, where image features are manually designed to locate and segment individual characters. Subsequently, character recognition is performed through feature classifiers based on machine learning, such as SVMs, random forests, etc. Traditional text recognition methods typically require substantial prior knowledge and can only be applied in environments with minimal interference, making them suitable for simple detection tasks only, which do not work well in the task of detecting and recognizing text on a wheel hub.

With the development of deep learning technology, deep neural networks are better able to extract image features of text, facilitating detection and recognition tasks in complex scenes. The text detection and recognition methods using deep neural networks are categorized into cascaded approaches and end-to-end approaches. The cascaded approach treats text detection [9,10,11,12,13,14,15,16] and text recognition [8,17,18,19] as two separate tasks, training a text detection network and a text recognition network independently, and then cascading the two networks to achieve the functionality of both detection and recognition. Due to the independent training of the text detection network and the text recognition network without any information exchange between them, the accuracy of the cascaded method tends to be relatively lower. In contrast, end-to-end methods [3,4,5,6,20,21] perform both detection and recognition simultaneously within a single network. These networks struggle to converge when trained on the wheel hub text images. because hub text exhibits low contrast and unclear engraving, which is difficult to detect and recognize the text at the same time.

To address the issues of low character contrast, unclear embossing, and diverse orientations in wheel hub character detection and recognition, this paper designed a wheel hub text detection and recognition model called HubNet and constructed a dataset to support the training of the model. Specifically, a wheel hub text dataset is constructed. The images in the dataset are collected from real industrial production line scenarios, featuring evenly distributed character categories and diverse orientations. Subsequently, based on existing natural scene text detection and recognition algorithms, we propose HubNet, a wheel hub character detection and recognition model utilizing global and local information. HubNet comprises a detection head, a recognition head, and a feature cross-fusion module. The detection head employs the state-of-the-art text detection network DBNet++ [22] as both the word detection branch and the character detection branch. Simultaneously, inspired by GLASS [23], a feature cross-fusion module is designed to integrate global and local features, thereby enhancing the accuracy of text recognition. The lightweight character recognition head is designed based on residual modules. The contributions of this paper are summarized as follows:We establish a wheel hub text dataset, including 446 images, 934 word instances, and 2947 character instances. The images in the dataset originate from authentic factory production line scenes with uniformly distributed character categories and diverse orientations.We propose HubNet, an end-to-end model for text detection and recognition utilizing both global and local features, combining directional information from global features and details from local features, which improves the model’s accuracy in text detection and recognition.

The remaining sections of this paper are structured as follows: Section 2 introduces the dataset constructed for this study, analyzes its characteristics, and presents the structure of HubNet; Section 3 details the experimental setup and validates the model through experimentation; Section 4 discusses the results and limitations of the proposed model.

## 2. Materials and Methods

### 2.1. Dataset

Currently, publicly available text detection and recognition datasets predominantly focus on text in natural scenarios [24,25,26,27], whereas text on a wheel hub exhibits notable differences in features compared with those in common environments. In support of text detection and recognition tasks for wheel hubs, we utilized line scan cameras to capture images of wheel hubs and, based on these images, constructed a wheel hub text dataset.

The wheel hub images in the dataset have a resolution of 5120 × 5120 pixels, as shown in the left image of Figure 1. Attempting to annotate these images directly would be quite challenging, and network training can be hindered by images with such high resolutions. To reduce the difficulty of dataset annotation, meet the requirements for network training, and ensure that any character can be fully displayed within a single image, we divide the original images into 11 × 11 images in an overlapping way, with a resolution of 512 × 512 pixels each. Images that turn out to be black after the division are discarded. Some of the images after division are shown in the right image of Figure 1. Part of the images in the dataset are shown in Figure 2.

The wheel hub text dataset produced for this paper exhibits the following characteristics and advantages:

**Data collected from real production line scenarios**: Unlike existing public datasets where images are sourced from natural scenarios, the hub dataset focuses on images captured specifically from industrial production line scenes. These images are gathered directly from authentic industrial production lines, offering essential data support for the task of text detection and recognition on wheel hubs under industrial conditions. The hub dataset comprises a total of 446 images, 934 word instances, and 2947 character instances, as detailed in Table 1.

**The orientation of the text in the wheel hub text dataset is more diverse**: Due to the fact that workpieces do not enter the production line at a fixed angle during the manufacturing process, the orientation of the workpieces varies in the images captured by the line scan camera on the production line, leading to diversity in the direction of the embossed characters on the workpieces. Several existing public datasets [24,25,26,27] focusing on natural scene text have already acknowledged the significance of text orientation in text detection and recognition tasks and have incorporated images with multi-directional text into their datasets. However, the orientation of the text in these existing datasets is still mostly concentrated within the range of −60° to +60°. The dataset constructed in this paper, targeting the true scenarios on the production line, features text orientations ranging from −180° to +180°, thus exhibiting a greater degree of directional diversity. The statistical chart is illustrated in Figure 3a.

**The wheel hub text dataset features annotations at both the word and character levels**: Most existing public datasets offer only word-level annotations, whereas recognition networks require character-level annotations. This discrepancy necessitates that state-of-the-art deep learning algorithms pre-train on synthetic datasets. The character-level annotations provided in the wheel hub text dataset enable deep learning models to be trained and fine-tuned at the character level. Additionally, within the word-level annotations, the distribution of aspect ratios is diverse, which is beneficial for detecting and recognizing characters with different aspect ratios on the wheel hub. This distribution is depicted in Figure 3b. Concurrently, in the character-level annotations, characters are evenly distributed across categories, avoiding the overrepresentation of any particular character type, as shown in Figure 3c.

### 2.2. Model

To detect and recognize wheel hub text in the context of an industrial production line scenario, this paper designs HubNet, a wheel hub text detection and recognition model using global and local features. The architecture of HubNet is illustrated in Figure 4.

HubNet consists of a backbone, a detection head, a feature cross-fusion module, and a recognition head. Its operational workflow is as follows: The network takes images containing wheel hub text as input. These images first pass through the backbone and obtain the feature map. The detection head yields word-level and character-level bounding polygons by using the feature map. and a global feature map. Subsequently, the character-level bounding polygons are used to crop the global feature map, obtaining the global feature $F_{g}$. The original images are cropped according to these bounding polygons and fed into the recognition feature extractor, producing the local feature $F_{l}$. Both $F_{g}$ and $F_{l}$ are then fed into the feature cross-fusion module, generating the fused feature $F_{f}$. Ultimately, $F_{f}$ serves as the input to the recognition head, leading to the classification of each character. The output of the network comprises the locations of bounding polygons for words in the image and the category of each character.

#### 2.2.1. Detection Head

Word-level detection should be coupled with recognition methods built on word-level optimization, which has a significantly larger search space than direct character classification [6]. This inevitably makes the subsequent recognition head more complicated and difficult to train by requiring a longer training time with a larger amount of training samples. Besides, though character-level detection can make the subsequent recognition task easier by dealing with it as a character classification task, it treats characters individually, raising the issue of how to determine which characters belong to the same word.

Therefore, the detection head designed in this paper is bifurcated into a word detection branch and a character detection branch, with both adopting the detector of the DBNet++ [22] model, which predicts the mask as well as the boundary of the text by using differentiable binarization. The detection head can predict the bounding polygons [28] of words and characters at the same time, while the recognition head can use the patches cropped by character polygons to classify the category of each character, and through the positioning of words and characters, it can be quite straightforward to determine which word each character belongs to.

In details, the word detection branch and the character detection branch have the same architecture but are trained on word level and character level, respectively. Both branches predict the probability map *P* and the threshold map *T*, and an approximate binary map $\hat{B}$ is calculated by *P* and *F* using the following formula, where (i,j) indicates the coordinate point in the map: (1)$${\hat{B}}_{i,j}=\frac{1}{1+e^{-k(P_{i,j}-T_{i,j})}},$$
where *k* indicates the amplifying factor and it is set to 50 empirically. The differentiable binarization can not only help differentiate text regions from the background but also separate text instances that are closely joined [22].

#### 2.2.2. Feature Cross-Fusion

There are currently two mainstream approaches for text recognition: The first is a modular cascaded method, as illustrated in Figure 5a, which cascades independent text detection and recognition models. The recognition model uses uniformly aligned and resized text crops as inputs in the vertical direction, abstracting away issues of scale and rotation. The second approach is a combined end-to-end (E2E) architecture, depicted in Figure 5b, which adds a recognition head that directly operates on the latent features of the detection model. Feature sampling replaces cropping, allowing for joint end-to-end training of detection and recognition.

As the E2E architecture has increasingly become the dominant structure, scale, and rotation-invariant image cropping has been supplanted by feature sampling, with sampled features becoming highly sensitive to both scale and rotation. While the joint optimization approach in E2E architectures has improved recognition performance for words that are not of extreme sizes or orientations, words at scale extremes and those with severe rotations tend to be poorly recognized within the E2E framework [23].

In order to leverage the strengths of both architectures, this paper proposes a feature cross-fusion module, as illustrated in Figure 6.

The feature cross-fusion module takes the global feature $F_{g}$ and the local feature $F_{l}$ as inputs; first, it concatenates $F_{g}$ and $F_{l}$ to obtain $f_{concat}$ with a shape of (2C,W,H).
(2)$$f_{concat}=concat\left(F_{g},F_{l}\right),$$

Subsequently generate the attention vector $y\in R^{2C}$, as shown in the formula below: (3)$$y=vec{\left(f_{concat}\right)}^{T}\times vec\left(v^{T}f_{concat}\right).$$

Among them, $v\in R^{2C}$ is a learnable vector, and the vec() operation transforms a tensor of shape (C, W, H) into a matrix of shape (C, WH). The fused feature map is ultimately generated using the attention vector *y*, as shown in the formula below: (4)$$F_{f}=f_{concat}+y^{T}f_{concat}.$$

Ultimately, the fused feature map $F_{f}$ is fed into the recognition head to classify the characters in the image.

The feature cross-fusion module overcomes the drawbacks of both the cascaded text detection and recognition network (where the text detection network and the text recognition network are trained separately) and the end-to-end text detection and recognition network (where both networks are trained simultaneously), thereby enhancing the accuracy of character recognition.

#### 2.2.3. Recognition Head

The character recognition algorithm designed in this paper adopts ResNet [29] as its base, taking the feature map $F_{f}\in R^{C\times H\times W}$ as input, where the C is 256. This results in a classification outcome with 37 categories (1 background + 10 Arabic numerals + 26 alphabetic characters). The specific model of the network is illustrated in Figure 7.

The feature map $F_{f}\in R^{C\times H\times W}$, obtained after feature fusion, is first fed into four cascaded ResBlock modules. Following this, average pooling is applied to the feature map, resulting in a vector $F_{p}\in R^{C\times 1\times 1}$. This vector $F_{p}\in R^{C\times 1\times 1}$ is then flattened and passed through a Multi-Layer Perceptron (MLP), yielding probabilities for 37 classes. Consequently, the final outcome of character classification is achieved.

#### 2.2.4. Loss Function

The training of the wheel hub text detection and recognition network using global and local features designed in this chapter is divided into two sections: the detection head and the recognition head. The detection head and the recognition head are trained separately.

For the detection head, following [22], the loss function $L_{d}$ can be expressed as the weighted sum of the loss of the probability map $L_{s}$, the loss of the binary map $L_{b}$, and the loss of the threshold map $L_{t}$.
(5)$$L_{d}=L_{s}+\alpha L_{b}+\beta L_{t}$$

Based on the values dictated by the loss function, the parameters α and β are respectively set to 1.0 and 1.0. Here, $L_{s}$ and $L_{b}$ denote the Binary Cross-Entropy (BCE) loss functions. $L_{t}$ represents the sum of L1 distances between predictions and labels within the dilated text polygon.

For the recognition head, the loss function $L_{r}$ employs the multi-class cross-entropy loss, defined as follows: where x represents the predicted probabilities, *y* is the ground truth label, *w* is the weight, *C* denotes the number of classes, and *N* is the number of samples in a batch: (6)$$L_{r}=-\frac{1}{N}\sum_{n=1}^{N}\sum_{c=1}^{C}w_{c}\log\frac{exp(x_{n,c})}{\sum_{i=1}^{C}exp\left(x_{n,c}\right)}y_{n,c}$$

## 3. Experiment

### 3.1. Experimental Environment Configuration

Compared to traditional character detection and recognition methods, artificial intelligence and deep learning possess a larger parameter space and superior feature learning capabilities. Upon entering the era of big data, the deep learning process involves massive matrix multiplications and intricate non-linear computations, demanding hardware with substantial computational power and adequate storage capacity. We train and test the HubNet on an NVIDIA GeForce RTX 3090 GPU with 24 GB of memory. The experimental environment and hardware configuration used in this paper are detailed in Table 2.

In the process of developing the proposed model, this paper initially trains the detection head and the recognition head separately. Subsequently, both branches are integrated and undergo fine-tuning together.

For the detection head, we first utilize pre-trained weights obtained from 100,000 iterations of pre-training on the SynthText dataset. Following this, we fine-tune the detection head for 1200 epochs on the wheel hub text dataset constructed in this paper, with the batch size set to 16. We adopt a poly learning rate strategy, where the current iteration’s learning rate equals the initial learning rate multiplied by ${\left(1-\frac{iter}{max\_iter}\right)}^{power}$, with the initial learning rate set to 0.007 and power equal to 0.9. We apply a weight decay of 0.0001 and momentum of 0.9. max_iter represents the maximum number of iterations. Data augmentation for the training data includes random rotation with an angular range of (−10°, 10°), random cropping, and random flipping. All processed images are resized to 640 × 640 to enhance training efficiency.

For the recognition head, network parameters are initialized using Kaiming initialization [30], followed by 50 rounds of training on the hub character dataset constructed for this paper. The batch size for training is set to 64, with the input being the feature maps obtained from the backbone network and the final prediction being probability values for 37 classes. During the training of the recognition head, the initial learning rate is set to 0.001, and after every 20 rounds of training, the learning rate is reduced to one-tenth of its previous value.Data augmentation for the training data includes random offset of the bounding boxes with a range of (−5, 5) pixels, as well as random rotation with an angular range of (−180°, 180°). Throughout the training process of the detection head, no stretching operations are performed on the cropped feature maps to ensure that the details within the feature maps remain unchanged.

Finally, the pre-trained detection head and recognition head are combined for joint training. The number of training rounds is set to 50, with a learning rate of 0.0001. After every 20 rounds, the learning rate is decreased to 0.9 times the current learning rate. At this stage, data augmentation is applied to the input images, including random rotation within an angular range of (−10°, 10°), random cropping, random flipping, random lighting variations, Gaussian noise, and salt-and-pepper noise. The loss and accuracy curves of the training process are shown in Figure 8.

### 3.2. Analysis on Backbone and Feature Extractor

Deformable convolutions [31] provide flexible receptive fields for the backbone network with very little extra computational cost, especially advantageous for targets with large aspect ratios, such as word-level detections in text detection, and DBNet++ [22] improves model performance by adopting deformable convolutions. In the experiment, we also consider EfficientNet [32] as one of the candidate backbone networks. For the feature extractor, we conducted experiments with standard ResNet-18 and ResNet-18 that utilize deformable convolutions. Through experimentation on the dataset constructed in Section 2.1, it is observed, as shown in Table 3, that incorporating deformable convolutions boosts word-level detection accuracy by 4.3%, while the effect on character-level detection accuracy is not significantly influenced by whether the backbone network uses deformable convolutions. Subsequent ablation studies are carried out on a backbone network composed of a ResNet-50 with deformable convolutions, alongside a feature extractor based on a ResNet-18 utilizing standard convolutions.

### 3.3. Ablation Study on Feature Cross-Fusion Module

To validate the role of the feature cross-fusion module, we replace it with alternatives that use only global features, only local features, concatenate global and local features, or add global and local features together. As shown in Table 4, the feature cross-fusion module outperforms these alternatives by increasing recognition precision by 15.6%, 3.1%, 2.6%, and 1.7%, respectively. Further analysis reveals that using only global features for character recognition yields the lowest accuracy, since global features derived from the backbone network are more conducive to the binary classification task of text detection rather than the multiclass classification task of text recognition involving 37 categories. Using only local features for recognition neglects directional or contextual information present in global features, leading to a decrease in recognition success. Similarly, simply concatenating or summing global and local features does not match the performance of the feature cross-fusion module due to the mismatch in scale between global and local features, which requires harmonization through the feature cross-fusion process. Visualizations of the experiment results are displayed in Figure 9.

### 3.4. Quantitative Comparison

We compared withpare the proposed HubNet against four representative baselines, including SPTS [20], SwinTextSpotter [4], Mask TextSpotter v3 [3], and ABCNet [33]. We conduct the comparisons with other approaches on both detection and recognition at the word level. For a fair comparison, all methods are under the same train/test split. Experiment results are shown in Table 5, where we can see that HubNet outperforms the current methods on the Wheel Hub Text Dataset.

Networks such as ABCNet and MaskTextSpotter v3 perform poorly on the dataset. According to our analysis, the reason is that these networks attempt to detect and recognize hub text simultaneously, and unclearly embossed text makes it difficult for the networks to converge, thus leading to poorer performance. Due to the failure to integrate global and local information, SwinTextSpotter and SPTS v2 also exhibit poor recognition performance on texts with extreme orientations and unclear embossing.

## 4. Discussion

To deal with the challenge of automatically detecting and recognizing the text on the wheel hub, we set up a dataset consisting of wheel hub images, which do good for the search of text detection and recognition, and it can also be a completion of a natural scene text dataset.

HubNet outperforms existing models on the wheel dataset. The feature cross-fusion module combines global features with local features, improving the accuracy of text detection and recognition and allowing the model to successfully detect and recognize text with extreme orientations and unclear imprints, proving the significance of global and local features in text detection and recognition tasks.

However, despite surpassing current models, the recognition accuracy of HubNet remains contingent upon the clarity of the imprinted characters on the wheel hub. Moreover, the training methods of current models are complex, and cascaded training may lead to a decline in model accuracy. Future work will involve refining this model by exploring more advanced networks for the recognition head and adopting an end-to-end training approach, which may further elevate model accuracy, enabling efficient and precise end-to-end detection and recognition.

## 5. Conclusions

Compared to natural scene texts, characters on wheel hub present challenges due to erosion, low contrast, and varied orientations, making automated detection and recognition particularly difficult. To address these issues, a wheel hub text dataset was constructed, featuring a diverse distribution of character orientations and word aspect ratios with a relatively balanced distribution of character label categories, providing the data foundation for the research. Moreover, we proposed HubNet, a wheel hub text detection and recognition model consisting of a detection head and recognition head conventionally, and a feature cross-fusion module proposed in this paper, which utilizes global and local features simultaneously to improve the recognition accuracy. Experiments conducted on HubNet have shown the effectiveness of the model and the feature cross-fusion module, which achieves a favorable outcome on the wheel hub text dataset.

## Figures and Tables

**Figure 1 sensors-24-06183-f001:**
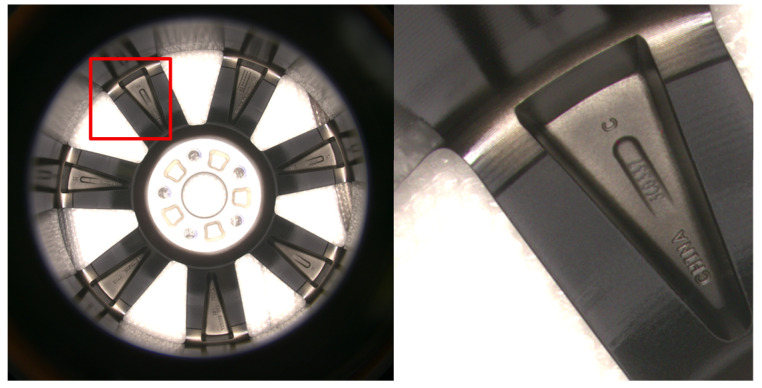
Collected wheel hub images (**left**) and image cropped at the red box (**right**).

**Figure 2 sensors-24-06183-f002:**
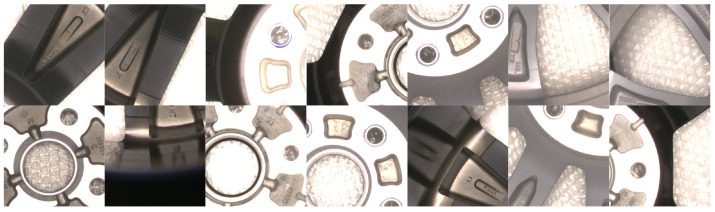
Part of the images in the dataset.

**Figure 3 sensors-24-06183-f003:**
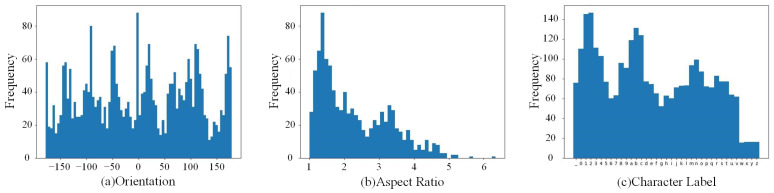
Analysis of Wheel Hub Text Dataset. (**a**) Distribution of character orientation in the dataset. (**b**) Distribution of word aspect ratio in the dataset. (**c**) Distribution of character labels in the dataset.

**Figure 4 sensors-24-06183-f004:**
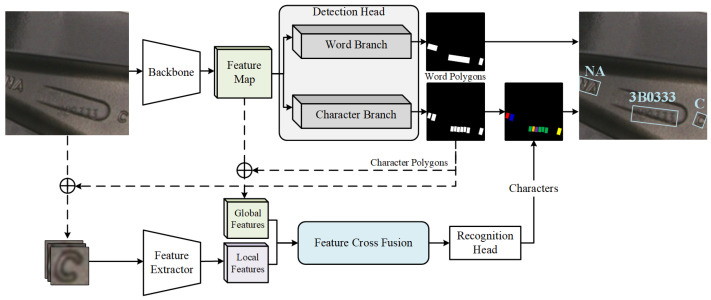
Structure of HubNet. Images are fed through a backbone network to obtain feature maps, upon which word and character detection is performed. Following character detection, the feature maps are cropped to derive global features, and the cropped images are passed through a feature extractor for local feature acquisition. These global and local features undergo the feature cross-fusion module, after which the features are input into a recognition head to yield character recognition results, thus completing the task of wheel hub character detection and recognition.

**Figure 5 sensors-24-06183-f005:**
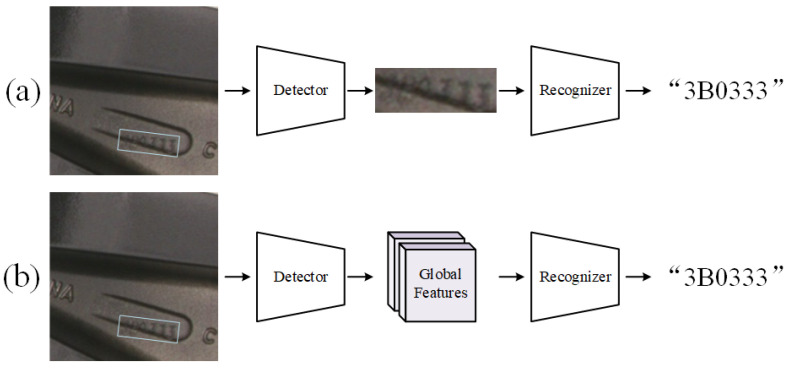
Two existing text detection and recognition frameworks. (**a**) Using local features only for text recognition. (**b**) Using global features only for text recognition.

**Figure 6 sensors-24-06183-f006:**
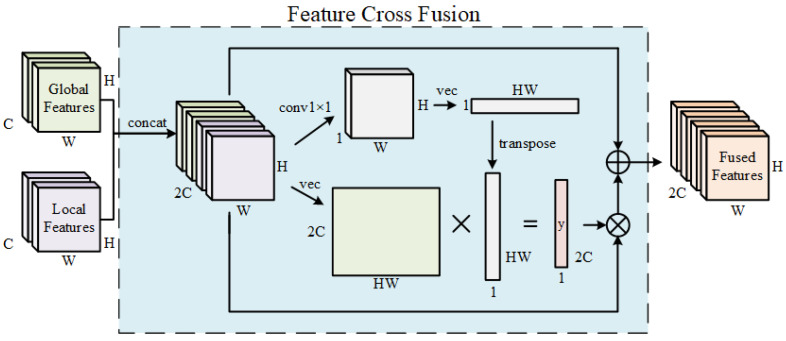
Structure of the feature cross-fusion module.

**Figure 7 sensors-24-06183-f007:**
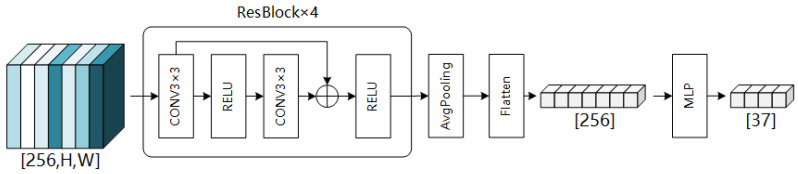
Structure of the recognition head.

**Figure 8 sensors-24-06183-f008:**
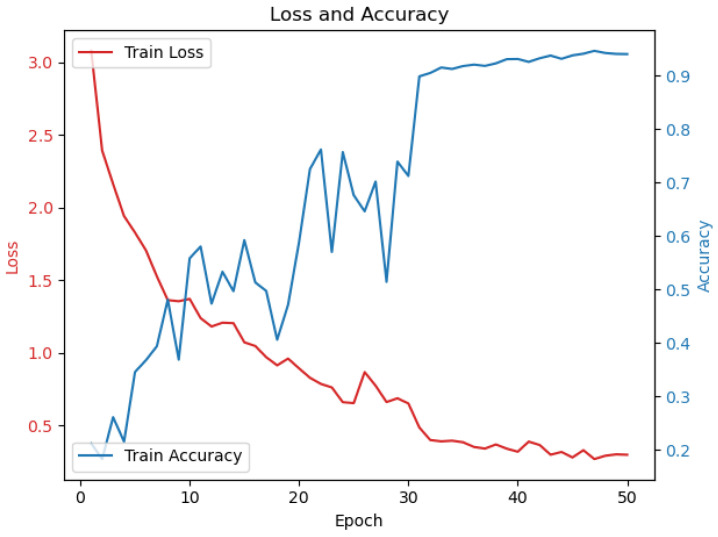
Loss and accuracy curves during training.

**Figure 9 sensors-24-06183-f009:**
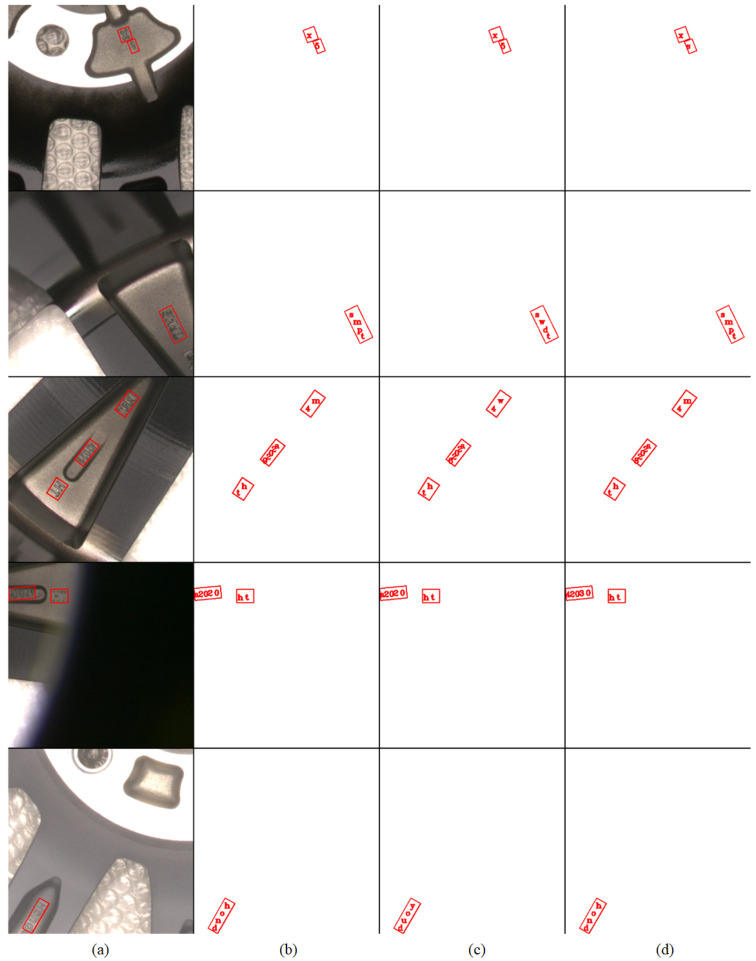
Visualization results of ablation studies on wheel hub text images. (**a**) Original Wheel Hub text images with location of text. (**b**) Proposed Model. (**c**) Model without FCF, using local features only. (**d**) Model without FCF, using global features only.

**Table 1 sensors-24-06183-t001:** Dataset Statistics.

Project	Quantity
Image resolution	512 × 512 pixels
Number of word instances in the dataset	934
Number of character instances in the dataset	2947

**Table 2 sensors-24-06183-t002:** Configuration of our experiment.

Software/Hardware	Version
CPU	Intel® Core™ i7-14700KF
GPU	NVIDIA GeForce RTX 3090
Operation System	Ubuntu 20.04
Python Environment	Conda 24.12
Python version	Python3.8.19
Pytorch version	Pytorch 1.10.1 + CUDA11.3
MMEngine version	0.10.3
MMDetection version	3.1.0
MMOCR version	1.0.1

**Table 3 sensors-24-06183-t003:** Analysis on backbone and feature extractor (FE).

Backbone	FE	Detection			Recognition		
		Presicion	Recall	F1	Precision	Recall	F1
EfficientNet	ResNet18D	90.7	89.5	90.1	80.8	73.4	76.9
EfficientNet	ResNet18	90.7	89.5	90.1	82.0	73.6	77.6
ResNet50	ResNet18D	88.1	80.9	84.3	81.3	72.2	76.5
ResNet50	ResNet18	88.1	80.9	84.3	80.0	70.8	75.1
ResNet50D	ResNet18D	**92.4**	**90.8**	**91.6**	85.5	78.7	82.0
ResNet50D	ResNet18	**92.4**	**90.8**	**91.6**	**86.5**	**79.4**	**82.8**

The bold numbers represent the best experimental results.

**Table 4 sensors-24-06183-t004:** Ablation study on the proposed feature cross-fusion module.

Features	Recognition		
	Precision	Recall	F1
Global only	70.9	64.4	67.5
Local only	83.4	75.2	79.0
Add	83.9	75.1	79.3
Concat	84.8	77.6	81.0
Feature cross-fusion	**86.5**	**79.4**	**82.8**

The bold numbers represent the best experimental results.

**Table 5 sensors-24-06183-t005:** Quantitative comparison of current methods.

Model	Venue	Detection			Recognition		
		Precision	Recall	F1	Precision	Recall	F1
Mask TextSpotter v3	ECCV’2020	82.4	80.8	81.6	33.1	30.5	31.7
ABCNet	CVPR’2020	56.5	57.3	56.9	12.4	10.9	11.6
SwinTextSpotter	CVPR’2022	91.4	90.5	90.9	74.1	69.3	71.6
SPTS v2	TPAMI’2023	91.2	**91.8**	91.5	79.2	72.7	75.8
Ours	–	**92.4**	90.8	**91.6**	**86.5**	**79.4**	**82.8**

The bold numbers represent the best experimental results.

## Data Availability

Data are available at https://github.com/Zeng1376/WheelHubTextDetectionRecognition, accessed on 1 January 2024.

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
