# Peer review of "HubNet: An E2E Model for Wheel Hub Text Detection and Recognition Using Global and Local Features"

_sensors, 2024, doi:10.3390/s24196183_

Round 1

Reviewer 1 Report

Comments and Suggestions for Authors

Paper Title:   HubNet: An E2E Model for Wheel Hub Text Detection and Recognition Using Global and Local Features

The authors of this paper construct a wheel hub text dataset, and proposes a wheel hub text detection and recognition model called HubNet.

  Here are some overall remarks.

Overall evaluation:

- The topic of the paper is very interesting (Deep Learning: automatic detection and recognition of wheel hub text)

Organization:

1- The paper is well organized, and the ideas are well structured.

GRAMMAR AND STYLE

1- Some grammatical errors should be corrected.

2- The author's writing style is clear.

3- Paragraphs and sentences are consistent.

CITATIONS

1- The references contain the recent works.

QUESTIONS

1.     Revise the abstract of the paper because it does not accurately reflect the content of the work.

 2.     In the abstract, the objective of this work must be described briefly with data in order to show the effectiveness of the review.

 3.     The authors need to specify whether the training was performed using a CPU or GPU, and the specific characteristics should be clearly detailed.

 4.     If possible, to add the curves of training Accuracy and loss for all HubNet architecture.

 5.     For the evaluation should, add a paragraph for the test and evaluation the performance of all the approaches.

 6.     The authors did not clearly articulate the objective of the paper.

 7.     For the evaluation should, add a paragraph for evaluation the performance of the approach like confusion matrix, reports classifications of models?

Comments on the Quality of English Language

Minor editing of English language required.

Reviewer 2 Report

Comments and Suggestions for Authors

The authors propose a novel dataset specifically for wheel hub analysis, which appears to be well-designed and carefully curated.

The proposed model is logical and seems appropriate for the task at hand.

The training workflow appears to be mostly sound and well-structured.

However, the authors should consider incorporating random brightness and contrast adjustments for data augmentation. These are simple yet effective transformations; it would be helpful to clarify why they were not included.

Additionally, the authors should explore more backbone options beyond ResNet. Why was ResNet chosen specifically? For instance, EfficientNet has been shown to offer better performance in some cases and could be a valuable comparison.

The authors should provide more details regarding the limitations of the proposed approach to offer a more balanced perspective.

Comments on the Quality of English Language

The manuscript would benefit from a thorough review of the English language. A professional proofreading service could help enhance clarity and readability.

Reviewer 3 Report

Comments and Suggestions for Authors

I would like to thank the author, I have just two commonets:

1- Revise the order and missing references.

2- Plagiarism need to be decreased.

Round 2

Reviewer 1 Report

Comments and Suggestions for Authors

the authors respected all the reviewers' comments.

Comments on the Quality of English Language

Minor editing of English language required.

Reviewer 3 Report

Comments and Suggestions for Authors

Best of Luck